# 🐂 STEER: Unified Style Transfer with Expert Reinforcement

**Skyler Hallinan**♡  **Faeze Brahman**♡♣  **Ximing Lu**♡♣
**Jaehun Jung**♡  **Sean Welleck**♡♣◇  **Yejin Choi**♡♣

♡Paul G. Allen School of Computer Science & Engineering, University of Washington
♣Allen Institute for AI  ◇Language Technologies Institute, Carnegie Mellon University
hallisky@cs.washington.edu

## Abstract

While text style transfer has many applications across natural language processing, the core premise of transferring from a single source style is unrealistic in a real-world setting. In this work, we focus on arbitrary style transfer: rewriting a text from an *arbitrary, unknown* style to a target style.

We propose STEER: **Unified Style Transfer with Expert Reinforcement**, a *unified* framework developed to overcome the challenge of limited parallel data for style transfer. STEER involves automatically generating a corpus of style-transfer pairs using a product of experts during decoding. The generated offline data is then used to pre-train an initial policy before switching to online, off-policy reinforcement learning for further improvements via fine-grained reward signals. STEER is unified and can transfer to multiple target styles from an arbitrary, unknown source style, making it particularly flexible and efficient.

Experimental results on a challenging dataset with text from a diverse set of styles demonstrate state-of-the-art results compared to competitive baselines. Remarkably, STEER outperforms the 175B parameter instruction-tuned GPT-3 on overall style transfer quality, despite being 226 times smaller in size. We also show STEER is robust, maintaining its style transfer capabilities on out-of-domain data, and surpassing nearly all baselines across various styles. The success of our method highlights the potential of RL algorithms when augmented with controllable decoding to overcome the challenge of limited data supervision.[1]

## 1   Introduction

Style transfer has been widely explored in the NLP field due to its practical applications, such as making text more formal (Rao and Tetreault, 2018),

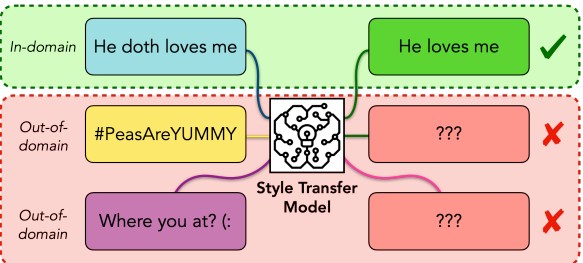

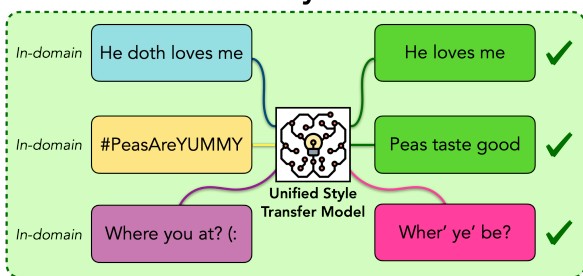

Figure 1: An overview of unified style transfer. In standard style transfer, models can only transfer from a single source style to a specified target style, struggling to transfer from out-of-domain texts. In contrast, **unified** style transfer models can transfer from an *arbitrary* source style to *multiple* target styles.

increasing politeness (Madaan et al., 2020; Mukherjee et al., 2023), or anonymizing authorship (Shetty et al., 2017; Patel et al., 2022). Previous work has mostly focused on one-to-one style transfer which involves rewriting text from one specific style to another while preserving meaning and fluency (Li et al., 2018; Sudhakar et al., 2019; Shen et al., 2017a). However, this approach may be less practical in real-world scenarios, where there are multiple and often unknown source styles a user wishes to transfer from.

We focus on **arbitrary style transfer**, a *many-to-one* style transfer task, where the goal is to transfer text from an arbitrary, unknown style to a target style using a single model (Reif et al., 2021; Krishna et al., 2020). This is a challenging task mainly

---

[1]We release our code publicly at https://github.com/shallinan1/STEERStyleTransfer

due to the lack of large-scale, human-curated corpora for training. Furthermore, we design a framework for training a **unified**, *many-to-many* style transfer model, which can do arbitrary style transfer to *multiple* target styles, as shown in Figure 1. To circumvent the lack of supervised data, recent approaches (Suzgun et al., 2022; Patel et al., 2022) heavily rely on large language models like GPT-Neo (Black et al., 2022) and GPT-3 (Brown et al., 2020) in zero or few-shot settings. Though promising and convenient, these approaches are limited by the high cost of API calls (OpenAI, 2023) and lack of reproducibility due to over-reliance on LLMs (Dean, 2023). Our method enhances the effectiveness of smaller, more accessible models for style transfer, broadening their adaptability and utility for the wider community.

In this work, we present **Unified Style Transfer with Expert Reinforcement ( 🐏 STEER)**, a novel, unified framework for many-to-one style transfer without supervision. Starting with a non-parallel corpus of text with various styles and a general paraphraser model, STEER first creates a diverse, pseudo-parallel dataset of style transfer pairs using product-of-experts decoding (Hinton, 2002; Liu et al., 2021). This makes our framework efficient by eliminating the need for costly human-curated datasets. Next, STEER uses offline reinforcement learning (RL) with this data before switching to online, off-policy RL for further improvement. To reflect the varied properties of style transfer, we adapt the QUARK algorithm (Lu et al., 2022), incorporating multiple reward models associated with different aspects such as style strength, fluency, and meaning similarity. Our framework is both practical and flexible, enabling a single model to transfer *arbitrary* source styles to *multiple* target styles.

We apply STEER to a diverse dataset of 11 styles (Krishna et al., 2020), developing a unified style model capable of transferring text from any of the 11 styles to any other style in the corpus. Our final model is effective at transferring style while preserving fluency and semantic similarity for all source and target styles, beating strong baselines across a suite of automatic metrics for style transfer. In particular, across all styles our 775M parameter model beats all baselines in overall style transfer quality, including the instruction-tuned 175B parameter GPT-3 model (Ouyang et al., 2022). Finally, we showcase the robustness of our model through evaluation on two out-of-domain source

styles that are unseen during training, where STEER consistently outperforms almost all baselines for every target style. The success of STEER demonstrates the effectiveness of reinforcement learning abetted by a high-quality, offline dataset in lieu of a good initial policy.

## 2 Task: Unified Style Transfer

Conventionally, the goal of style transfer is to take an input text in a known source style $\mathbf{x_{s_i}}$ and rewrite it into some known target style $\mathbf{x}_{s_j}$ while preserving meaning and fluency. However, this setting is unrealistic and may not cover real-world use cases where there are multiple and often unknown source styles. The goal of **arbitrary style transfer** is to instead transfer text from an *arbitrary, unknown* style to a text in the target style with meaning and fluency preservation. Formally, given $\mathcal{S}$ as the set of all possible style choices, this amounts to finding a function $f : \mathcal{X} \times \mathcal{S} \to \mathcal{X}$, which takes an input text $\mathbf{x}$ and a desired target style $s_j$, and outputs a modified text in the target style $\mathbf{x_{s_j}}$.

## 3 Unified Style Transfer with Expert Reinforcement

We introduce STEER, a novel two-stage framework for unsupervised unified style transfer. Our framework is illustrated in Figure 2 and is composed of 1) **expert-guided data generation** to circumvent the challenge of obtaining supervised datasets at scale, and 2) **offline reinforcement learning** followed by **online reinforcement learning** to effectively align an initial policy with multiple reward functions related to the style transfer task.

In *expert-guided data generation* (§3.1), the goal is to automatically collect a diverse high-quality dataset $\mathcal{D}_f$ of style transfer pairs using only a general paraphraser $\mathcal{M}_p$ and a corpus of diverse styles $\mathcal{C}$. To this end, we follow an *overgenerate-and-filter* approach: we first generate a large pool of candidate pairs from the paraphraser guided by style expert models in a product-of-expert fashion (Hinton, 2002), then leave only pairs that qualify for the style transfer task (i.e., accurately transferred style and semantically similar pairs). In *online off-policy reinforcement learning* (§3.2), we first update the paraphraser $\mathcal{M}_p$ as an initial policy using supervised learning on the collected dataset and then switch to online, off-policy learning for further data exploration and model improvements (Ramamurthy et al., 2022; Lu et al., 2022).

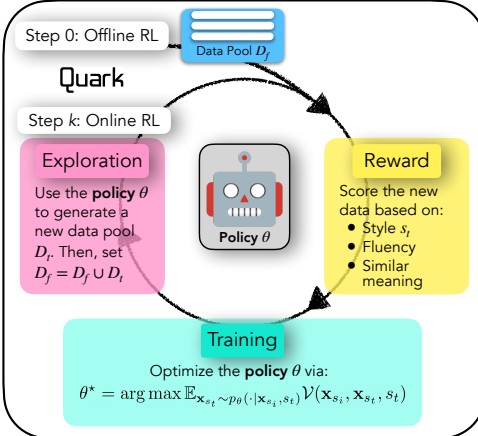

**(1) Expert-guided Data Generation**

**(2) Reinforcement Learning**

Figure 2: An overview of STEER. We first use **expert-guided data generation** to automatically generate candidate style-transfer pairs $\mathbf{x}_{s_i} \to \tilde{\mathbf{x}}_{s_t}$, mapping from an input of arbitrary style $x_{s_i}$ to a rewrite $x_{s_t}$ in a target style, by decoding with a product of experts using a paraphraser $\mathcal{M}_P$ and style-expert LMs. After filtering by quality metrics, we have a diverse, high-quality dataset $\mathcal{D}_f$. We then train a unified, *many-to-many* style transfer model, using $\mathcal{D}_f$ for **offline RL** before switching to **online, off-policy RL** to further optimize style transfer quality.

## 3.1 Expert-guided Data Generation

We first leverage expert LMs to generate a high-quality, pseudo-parallel style transfer corpus.

**Generation** For each target style $s_t \in \mathcal{S}$, we first massively *generate* a diverse set of *candidate* style transfer pairs $\mathbf{x}_{s_i} \to \tilde{\mathbf{x}}_{s_t}$ for all $s_i \in \mathcal{S} - \{s_t\}$, such that we collect pairs of transfers from each possible source style to the target style. To do so, we first-pass text $\mathbf{x}_{s_i}$ from a candidate source style through a general (style-agnostic) paraphraser $\mathcal{M}_P$, typically resulting in a normalized text $\tilde{\mathbf{x}} = \mathcal{M}_P(x_{s_i})$ with little or no stylistic features (Krishna et al., 2020). To ensure that the $\tilde{\mathbf{x}}$ belongs to the desired target style, we *steer* the paraphraser $\mathcal{M}_P$ generation towards the target style and away from the source style during decoding. Intuitively, we exploit the inherent capability of the paraphraser to faithfully rewrite input texts, while injecting stylistic control through guided-decoding.[2]

To do this, we leverage DEXPERTS decoding (Liu et al., 2021), a controllable text generation paradigm that enables steering towards and away from distinct attributes. DEXPERTS combines the distribution of a base autoregressive model $P_b$ with those of an "expert" $P_e$ and/or "anti-expert" $P_a$ model in a product of experts, which are trained on desirable and undesirable attributes respectively. Given a prompt $x_{<t}$, the next token probability is

obtained by a product-of-experts:

$$P(x_t|x_{<t}) \propto P_b(x_t|x_{<t})\left(\frac{P_e(x_t|x_{<t})}{P_a(x_t|x_{<t})}\right)^{\alpha} \quad (1)$$

where $\alpha$ is hyperparameter controlling the strength of control over the base model $P_b$.

Within our problem setting, we consider the general paraphraser $\mathcal{M}_p$ as the base model, and two language models finetuned on texts belonging to target style $s_t$ and source style $s_i$ as our expert and anti-expert models respectively. Given a text in a candidate source style $\mathbf{x}_{s_i}$, we generate text in the target style $\mathbf{x}_{s_t}$ via sampling from the probability distribution obtained in Eq. 1. We repeat this expert-guided decoding for all the source and target styles, resulting in a dataset $\mathcal{D}_{init}$. In practice, we *over*-generate data by repeating the generation procedure above with a vast sweep of hyperparameters, such as multiple sampling temperatures and decoding algorithms, so we can eventually filter and attain as many high-quality rewrites possible.

**Filtering** Not all of the expert-guided generations in $\mathcal{D}_{init}$ are high-quality. We thus *filter* $\mathcal{D}_{init}$ and retain the pairs that best represent the task of style transfer. We assess the quality of each candidate style transfer pair in $\mathcal{D}_{init}$ with three standard style transfer metrics:

1. **Target Style Strength (TSS)** of the generation $\mathbf{x}_{s_t}$ is measured by the probability of the target class $s_t$ with a RoBERTa-large classifier (Liu et al., 2019) trained on text from all

---

[2]In practice, this procedure can be repeated to add any new target style to the dataset.

the styles in the corpus $\mathcal{C}$. Both style strength and style accuracy have been used in previous work (Reif et al., 2021; Krishna et al., 2020); we opt for style *strength*, as it it is more fine-grained than a binary measurement of accuracy. Accordingly, we train our classifier in a *multi-label* setup, such that the prediction probability of each target style can be independently evaluated.

2. **Fluency (F)** of the generation $\mathbf{x}_{s_t}$ is measured by the probability of being grammatically acceptable via a binary RoBERTa-large classifier trained on the CoLA dataset (Warstadt et al., 2018).

3. **Meaning Similarity (MS)** between the input $\mathbf{x}_{s_i}$ and rewritten text $\mathbf{x}_{s_t}$ is measured via SentenceTransformers embedding distance (Reimers and Gurevych, 2019).

Following previous work (Krishna et al., 2020), for each candidate style transfer pair, we aggregate the three style metrics above into a joint metric $\mathcal{V}$ that captures the overall quality:

$$\mathcal{V}(x_{s_i}, x_{s_t}, s_t) = \mathbf{TSS}(x_{s_t}, s_t) \cdot \mathbf{F}(x_{s_t}) \cdot \mathbf{MS}(x_{s_i}, x_{s_t})$$

All three individual metrics are scalar values in the interval $[0, 1]$[3], which ensures also that $\mathcal{V} \in [0, 1]$.

Next, we filter our data to create a high-quality pool of training data $\mathcal{D}_f$ for subsequent model training. For each target style in $\mathcal{D}_{init}$, we sort the style-transfer pairs by their combined score $\mathcal{V}$, then take the top-$k$ examples. This sampling method ensures that the examples in the resulting dataset are the highest quality possible, but may also lead to lower diversity, as it excludes lower-scoring generations.

In practice, with multiple target styles in the initial pool of pairs $\mathcal{D}_{init}$, filtering is done for each style separately, and the filtered data from each target style is combined to form $\mathcal{D}_f$.

## 3.2 Reinforcement Learning

Next, we train a unified style transfer model by leveraging the generated corpus $\mathcal{D}_f$. Concretely, our goal is to attain a rewriting model $\mathcal{M}_\theta$ which accepts an input with arbitrary style $\mathbf{x}_{s_i}$ along with a target style $s_t$ and produces a high-quality rewrite $\mathbf{x}_{s_t}$, as evaluated by the joint metric $\mathcal{V}$, formally:

$$\theta^\star = \arg\max \mathbb{E}_{\mathbf{x}_{s_t} \sim p_\theta(\cdot | \mathbf{x}_{s_i}, s_t)} \mathcal{V}(\mathbf{x}_{s_i}, \mathbf{x}_{s_t}, s_t)$$

Recently, online policy-based RL algorithms (Lu et al., 2022; Schulman et al., 2017; Ramamurthy et al., 2022) have been shown effective in optimizing language models towards a given objective function. In the RL framework, we refer to the model $\mathcal{M}_\theta$ as the **policy** and the objective function $\mathcal{V}$ as the **reward**. Generally, online RL algorithms conduct policy optimization with model-generated outputs while assuming a reasonable degree of alignment between the output distribution of the initial policy and the optimal reward distribution. This alignment is necessary to produce generations with meaningful signals for RL training.

Due to the absence of supervision, the closest initial policy for our unified style transfer task would be the style-agnostic paraphraser $\mathcal{M}_P$. However, this initial policy is still far away from the optimal reward distribution as the style transfer task falls beyond the capabilities of the paraphraser $\mathcal{M}_P$, making it unable to produce useful generations for RL optimization. To overcome this challenge, we propose first conducting *offline* RL training and then progressing to *online* RL training. Specifically, prior to optimizing $\mathcal{M}_P$ with its own generations, we first perform RL optimization on the style transfer data $\mathcal{D}_f$ generated through expert-guidance (§3.1). Intuitively, the offline stage equips the initial policy with a certain degree of style transfer capability before online stage further optimizes it towards generating rewrites with better quality.

In practice, we employ and adapt the RL algorithm QUARK (Lu et al., 2022) to accomplish the two-stage RL training. QUARK is an online, off-policy RL algorithm that has proven effective in various text generation tasks. Notably, the off-policy nature[4] makes it possible to be adapted for the offline RL stage. QUARK optimizes a reward function through reward conditioning. Concretely, the algorithm alternates between 1) collecting samples with the current language model, 2) sorting them into quantiles based on their reward, with each quantile identified by a *reward token* prepended to the language model's input, and 3) using standard language modeling loss on samples from each quantile conditioned on their reward token.

When adapting QUARK to offline RL, we start by initializing the data pool with the style transfer corpus $D_f$ generated through expert-guidance rather than gathering generations from the initial

---

[3]SentenceTransformers occasionally outputs negative scores; we set these to 0 to ensure a score in $[0, 1]$

[4]Off-policy RL evaluates and improves a policy different from the policy used for action selection (i.e. data generation).

| Target Style | GPT-2 Large | | | GPT-3 (text-davincii-003) | | | |
|---|---|---|---|---|---|---|---|
| | **STEER** | STRAP | P-A-R | $k=0$ | $k=1$ | $k=5$ | $k=10$ |
| AAE Twitter | **42.6** | 7.4 | 3.8 | 23.2 | 11.2 | 25.4 | 22.7 |
| Bible | **44.0** | 26.9 | 6.6 | 5.2 | 16.0 | 20.2 | 21.0 |
| 1810-1820s | **30.2** | 11.1 | 3.5 | 14.7 | 15.9 | 17.4 | 17.0 |
| 1890-1900s | **35.9** | 12.3 | 4.4 | 8.6 | 9.1 | 10.4 | 10.1 |
| 1990-2000s | **42.3** | 16.6 | 4.3 | 7.9 | 13.0 | 17.5 | 17.2 |
| English Twitter | **41.2** | 8.0 | 5.5 | 35.0 | 23.6 | 32.0 | 29.5 |
| James Joyce | **20.4** | 11.8 | 5.4 | 3.4 | 1.3 | 1.6 | 2.6 |
| Song Lyrics | **33.3** | 20.2 | 7.7 | 12.2 | 15.4 | 11.2 | 13.2 |
| Romantic Poetry | **20.4** | 15.7 | 2.8 | 1.1 | 3.4 | 6.2 | 4.9 |
| Shakespeare | **13.6** | 9.1 | 2.5 | 9.6 | 10.0 | 9.7 | 9.7 |
| Switchboard | **52.9** | 21.1 | 1.7 | 0.1 | 0.3 | 5.3 | 13.7 |
| **Overall** | **34.3** | 14.6 | 4.4 | 11.0 | 10.8 | 14.3 | 14.7 |

Table 1: Comparison of 11-way style transfer on the CDS dataset measured by aggregate score $\mathcal{V}$ with different methods, including STRAP (Krishna et al., 2020) and P-A-R (Suzgun et al., 2022), using GPT-2 Large (774M), and GPT-3 (175B). **Bold** and underline denote the highest and the second-highest score respectively in each row.

policy. Afterward, we carry out the quantization and learning steps in the same manner as the original QUARK. After completing the offline RL stage, we proceed with the online QUARK training by alternating between data generation with the updated policy, quantization and learning. In both stages, our training objective can be written as:

$$\theta^{\star} = \max_{\theta} \mathbb{E}_{(\mathbf{x}_{s_i}, \mathbf{x}_{s_t}) \sim \mathcal{D}} \log p_{\theta}(\mathbf{x}_{s_t} | \mathbf{x}_{s_i}, s_t, r_{\mathcal{V}(\mathbf{x}_{s_i}, \mathbf{x}_{s_t}, s_t)})$$

where $r_{\mathcal{V}(\cdot)}$ denotes the quantized reward token corresponding to the reward score $\mathcal{V}(\cdot)$ of the generated rewrite. In online RL, $\mathcal{D}$ is expanded with samples from the improved policy at each iteration.

Additionally, we also explore integrating a vectorized reward function $\mathbf{v}(\mathbf{x}_{s_i}, \mathbf{x}_{s_t}, s_t)$ into the QUARK algorithm, rather than using the joint multiplied scalar score $\mathcal{V}$ as the reward function. In this case, instead of conditioning on one reward token that corresponds to a quantized scalar score, we condition on a reward vector composed of three reward tokens. These reward tokens represent quantized scores from the style, fluency and similarity metrics respectively. As we will show in the experiment section, we observe a noticeable performance boost brought by vectorized QUARK in terms of reward optimization. We believe this is likely because the vectorized reward provides additional fine-grained signals for optimization, which reflect the quality of each generated output with respect to individual evaluation metrics.

## 4 Experiments

We detail our experiment setup, including the datasets (§4.1), baselines (§4.2), evaluation metrics

(§4.3), experimental details (§4.4), main results (§4.5), ablations (§4.6), and analysis of $\mathcal{D}_f$ (§4.7).

### 4.1 Datasets

We use the following datasets in our experiments: 1) the **Corpus of Diverse Styles** (CDS; Krishna et al., 2020) is a non-parallel, diverse text corpus with 11 distinct styles such as Shakespeare and the Bible, 2) **Grammarly's Yahoo Answers Formality Corpus** (GYAFC; Rao and Tetreault, 2018) is a parallel corpus of formal and informal responses collected from the Yahoo Answers forum, and 3) the **Yelp Review Dataset** (Yelp; Shen et al., 2017a) is a non-parallel corpus of user-reviews on various businesses and services from the Yelp with binary sentiment ratings of positive or negative. For more details on the datasets see Appendix B.

### 4.2 Baselines

We use three competitive style-transfer baselines. Method-specific details are located in Appendix C:

**Style Transfer via Paraphrasing** (STRAP; Krishna et al., 2020) is an unsupervised approach for arbitrary style transfer, which uses GPT-2 Large (Radford et al., 2019) inverse paraphrasers.

**Prompt-and-Rerank** (P-A-R; Suzgun et al., 2022) prompts some language model to generate $k$ candidate style transfer texts, ranks them based on quality, and returns the best one. We use P-A-R with GPT-2 Large.

**GPT-3** (Brown et al., 2020; Ouyang et al., 2022) is a highly-capable class of decoder-only models,

| | GPT2-Large | | | | | | GPT-3 (`text-davincii-003`) | | | | | | | |
| | STEER | | STRAP | | P-A-R | | $k=0$ | | $k=1$ | | $k=5$ | | $k=10$ | |
| Target Style | Inf. | For. | Inf. | For. | Inf. | For. | Inf. | For. | Inf. | For. | Inf. | For. | Inf. | For. |
|---|---|---|---|---|---|---|---|---|---|---|---|---|---|---|
| AAE Twitter | **44.0** | **47.7** | 18.7 | 13.2 | 25.6 | 10.6 | 31.7 | 29.2 | 21.5 | 17.9 | 30 | 28.8 | 30.2 | 27.6 |
| Bible | **36.1** | **38.8** | 22 | 22.9 | 0.3 | 1.6 | 4.3 | 4.4 | 15.7 | 15.9 | 18.0 | 19.0 | 19.8 | 19.5 |
| 1810-1820s | **26.3** | **29.5** | 5.9 | 10.0 | 1.2 | 4.7 | 12.4 | 15.6 | 14.3 | 16.9 | 17.6 | 21.6 | 16.9 | 20.1 |
| 1890-1900s | **33.5** | **34.7** | 10.0 | 13.4 | 4.4 | 11.0 | 9.9 | 11.8 | 13.9 | 13.8 | 14.6 | 14.4 | 13.8 | 13.3 |
| 1990-200s | **50.2** | **56.2** | 22.6 | 32.1 | 11.8 | 31.4 | 16.7 | 20.7 | 28.5 | 32.5 | 31.5 | 34.7 | 28.4 | 32.8 |
| English Twitter | **46.1** | **54.1** | 20.1 | 22.1 | 32.4 | 33.5 | 37.4 | 41.8 | 30.1 | 29.5 | 34.9 | 36.4 | 32.5 | 35.0 |
| James Joyce | **22.3** | **22.8** | 10.9 | 13.2 | 3.2 | 7.9 | 2.9 | 3.3 | 2.7 | 2.3 | 3.1 | 2.5 | 3.3 | 2.8 |
| Song Lyrics | **42.6** | **40.5** | 22.1 | 23.2 | 10.3 | 12.4 | 19.3 | 12.9 | 22.3 | 18.4 | 19.3 | 16.2 | 24.2 | 20.1 |
| Romantic Poetry | **13.5** | **12.9** | 8.9 | 10.8 | 0.8 | 0.9 | 2.0 | 1.1 | 5.2 | 4.3 | 7.0 | 4.7 | 6.0 | 3.9 |
| Shakespeare | 11.8 | 11.6 | 11.1 | 10.4 | 1.3 | 4.1 | 12.9 | 15.1 | **15.3** | 14.7 | 13.4 | 15.2 | 13.8 | **15.2** |
| Switchboard | **54.6** | **59.3** | 29.7 | 35.1 | 5.2 | 6.1 | 0.1 | 0.1 | 0.3 | 0.1 | 9.7 | 13.4 | 15.6 | 23.0 |
| **Overall** | **34.6** | **37.1** | 16.5 | 18.8 | 8.8 | 11.3 | 13.6 | 14.2 | 15.4 | 15.1 | 18.1 | 18.8 | 18.6 | 19.4 |

Table 2: Comparison of style transfer to each of the 11 styles in the CDS dataset measured by aggregate score $\mathcal{V}$ from two out-of-domain styles from the GYAFC corpus. For. and Inf. denote the formal and informal styles respectively. **Bold** and underline denote the highest and the second-highest score respectively in each row.

particularly showing strong zero- and few-shot performance. We utilize GPT-3 as baseline both in a zero-shot and few-shot ($k = 1, 5, 10$) setting. Specifically, we use the instruction-tuned, 175B parameter engine `text-davinci-003`.[5]

### 4.3 Evaluation Metrics

To evaluate the quality of each style transfer pair, we use the same metrics introduced in §3.1: target style strength (TSS), fluency (F), meaning similarity (MS), and the aggregate metric $\mathcal{V}$. For a set of style transfer pairs (i.e., over an entire data corpus), we report the *average* $\mathcal{V}$.[6] To ensure that the improvement from STEER is meaningful (i.e., to make sure our model is not reward hacking), we also report evaluation using alternative metrics unseen during training in Appendix F; these results corroborate our main findings in §4.5.

### 4.4 Experimental Details

For all non-GPT-3 baselines, we use GPT-2 large as the base language model. Specifically, for STEER, we use GPT-2 large for the paraphraser and for the expert models. Our main STEER results are with the vectorized QUARK variant (i.e., using fine-grained reward). More details are in Appendix A.3.

### 4.5 Style Transfer on CDS

To evaluate STEER's capability on arbitrary style transfer, we use CDS, as it has 11 diverse styles.

Specifically, we train a *unified* model that can transfer arbitrary text to each style in the corpus $\mathcal{C}$. We use top-200$K$ filtering for each target style, resulting in $|\mathcal{D}_f| = 2.2$M. Finally, we evaluate style transfer to each target style by transferring from 1000 test-set examples from every other style $\in C$; this results in a total test-set size of 10,000 for each target style.

**Automatic Results** We demonstrate automatic results on the CDS in Table 1. Across all target styles, STEER outperforms all baselines on $\mathcal{V}$, the aggregate style transfer quality, including GPT-3, a model 226 times larger, and STRAP, a comparably-sized, non-unified baseline. This shows that with expert-guided data generation and offline-then-online RL, a *unified* model can outperform other models of the same or even much larger size. The full results, including individual style transfer metrics, are in Appendix G.1.

GPT-3 has its best relative performance on the Twitter and Shakespeare styles, but struggles otherwise. This shows the limitations of relying on large-scale general-purpose LLMs: in this case, GPT-3 excels transferring to styles most likely to be highly prevalent in it's internet text corpus (Brown et al., 2020) However, it is unlikely to generalize to more obscure styles unseen during training, even with few-shot examples. The poor performance of the GPT-2-based P-A-R reinforces this, showing the unreliability of prompting general-domain, pretrained LMs for style transfer, especially at smaller scales.

We also conduct an out-of-domain evaluation to assess the robustness of each method to unseen

---

[5] Most capable model during conducting this work.

[6] Though for each style transfer pair, $\mathcal{V}$ is equal to the product of TSS, F, and MS, once we take the corpus-average of these metrics we lose this equality guarantee

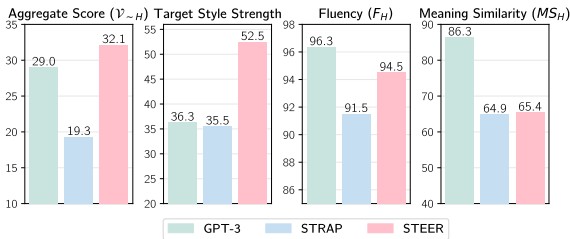

Figure 3: Style transfer quality $\mathcal{V}_{\sim H}$ on CDS, averaged across all 11 styles, with fluency and meaning similarity human evaluation. **TSS** is automatically computed.[10]

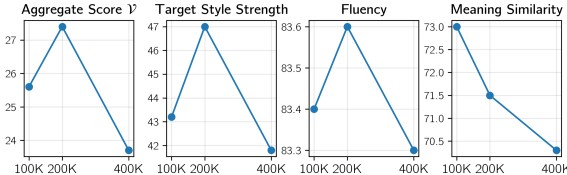

Figure 4: Plots of the style transfer quality on CDS averaged across all 11 styles with varying $k$, the hyper-parameter used in the Top$-k$ sampling strategy.

inputs. Specifically, we use text from the two styles in GYAFC as inputs at testing time,[7] employing the previously trained CDS model without further fine-tuning, and transfer to each of the 11 styles in CDS. Our results are shown in Table 2: overall, STEER is the most robust method, outperforming all others in total score, $\mathcal{V}$, for almost all target styles. STEER loses only to GPT-3 on the Shakespeare style; this may be due to the inherent knowledge of Shakespeare stored in GPT-3.

**Human Evaluation** We also conduct a human evaluation to verify the quality of the generations. We use a 3-point Likert Scale to evaluate style transfers (Iyyer et al., 2018) on both **meaning similarity** (MS$_H$) and **fluency** (F$_H$).[8] Afterwards, we scale MS$_H$ and F$_H$ down to $[0, 1]$ and multiply them by the automatically-computed **TSS** to attain $\mathcal{V}_{\sim H} \in [0, 1]$, a partially human aggregate metric. Three NLP experts annotate 10 examples per 11 target styles from 3 models (330 total examples).[9]

Figure 3 shows our human evaluation results. In terms of individual metrics, STEER has better fluency than STRAP and maintains competitive fluency to GPT-3, which is known to excel at generating human-like text (Brown et al., 2020). STEER also performs slightly better in meaning similarity than STRAP, but GPT-3 outperforms both of them significantly. However, the **TSS** of STEER makes up for this and dwarves both the baselines. We think this is a reasonable trade-off: STEER sacrifices much less fluency and meaning preservation for much more style transfer strength.

Previous work has also demonstrated this trade-

| Input: Can't sleep at all. Smh. | Transfer: AAE → 1990s-2000s |
|---|---|
| STEER | I mean, I can't sleep at all. |
| GPT-3 | I am unable to obtain any rest; shaking my head in disbelief. |
| STRAP | I don't want to sleep. |
| P-A-R | Can't sleep at all, smh. |
| **Input: Yeah one one way or another** | **Transfer: switchboard → rom. poetry** |
| STEER | One way, or another, or both |
| GPT-3 | Aye, one mayhap will find the way. |
| STRAP | And one way or the other |
| P-A-R | Yeah one one way or another |
| **Input: In his fear, he dare not face me** | **Transfer: lyrics → bible** |
| STEER | And he will not dare to face me: for fear of me is in his eyes. |
| GPT-3 | And his fear was great, so that he could not stand before me. |
| STRAP | For he that is afraid of me is of me; but he that is of me is of him. |
| P-A-R | In fear he came and hid himself, because God was near to him |

Table 3: Examples of style transfer pairs generated by STEER and other methods. GPT-3 is run with 10-shot.

off between style transfer accuracy and meaning preservation, both through empirical results (Suzgun et al., 2022; Malmi et al., 2020; Wu et al., 2019; Li et al., 2018) and explicit mentions in discussions (Li et al., 2018; Xu et al., 2019; Wu et al., 2019; Hallinan et al., 2023). Intuitively, when transferring from one style to another, some amount of semantic changes is unavoidable; as a simple example, meaning similarity will be maximized when the input is naively copied.

Overall, human evaluation validates our main findings: STEER still beats both baselines in overall score $\mathcal{V}$. These results show that GPT-3 is excellent at *paraphrasing* - creating fluent and semantically similar rewrites, but not at transferring to multiple diverse styles, as it often struggles to convert to the target style. On the other hand, STEER is more versatile, maintaining moderate-to-strong performance on all individual metrics, making it the strongest overall method.

Finally, we show qualitative examples of generations from different models in Table 3. In the examples, STEER produces style transfers that optimize across *all* dimensions, while other methods

---

[7]We use 1000 examples from each class

[8]As in Krishna et al., 2020, we do not conduct human evaluation for target style strength, as the task is to complex for untrained annotators unfamiliar with the target styles. Appendix D.1 details an experiment we conducted which verifies this task's hardness for annotators

[9]We omit P-A-R from human evaluation due to its low performance in automatic metrics. See Appendix D for details.

[10]Pairwise agreements are 94.8% and 98.8% for fluency and similarity respectively.

| Reward Type | $\mathcal{V}$ | TSS | F | MS |
|---|---|---|---|---|
| Coarse | 19.9 | 42.8 | 79.0 | 62.6 |
| Fine-grained | **27.4** | **47.0** | **83.6** | **71.5** |

Table 4: Style transfer quality on CDS, averaged across 11 target styles using STEER with a coarse vs a fine-grained reward. The highest values are denoted in **bold**.

| | | $n$-gram entropy, $n =$ | | | | |
|---|---|---|---|---|---|---|
| | MSTTR | 1 | 2 | 3 | F | MS |
| $\mathcal{D}_f$ (STEER) | 0.912 | 8.8 | 15.0 | 19.3 | 86.8 | 67.2 |
| GYAFC | 0.931 | 9.1 | 15.0 | 17.8 | 86.2 | 78.4 |
| Yelp | 0.958 | 9.6 | 16.4 | 21.3 | 87.2 | – |
| CDS | 0.946 | 10.0 | 17.0 | 20.4 | 83.2 | – |

Table 5: Data metrics on $\mathcal{D}_f$ (STEER) and other datasets.

optimize for only one or two.

## 4.6 Ablations

We perform two ablation studies to analyze the effect of dataset size and reward design in STEER. All models are compared after 15K training steps:

**Dataset Size** We investigate the effect of different dataset sizes on the performance of STEER. Using the top-$k$ sampling strategy, we vary $k$ with $k = 100K, 200K,$ and $400K$ and compare style transfer on CDS. Figure 4 shows the average results transferring to 11 target styles in CDS from all other styles.

Interestingly, we do not observe direct scaling of style transfer performance with increasing dataset size; as the top-$k$ value increases, the aggregate score $\mathcal{V}$, target style strength **TSS**, and fluency **F** all follow a *reverse* U-shape curve.

These results may indicate a trade-off between diversity and quality in the dataset $D_f$ used to train STEER: as the $k$ value increases with top-$k$ sampling, $D_f$ becomes more diverse, but also includes samples with lower-quality, which may hurt model performance downstream. On the other hand, when $k$ is too small, though the average quality of each example in $\mathcal{D}_f$ is higher, fewer diverse examples may hurt generalization. The optimal dataset has examples with sufficient *variety* and *quality*, enabling the model to learn a high-quality policy while staying resilient to various inputs.

**Coarse vs Fine-grained Reward** We also directly compare the use of coarse or fine-grained reward tokens in the RL stages of STEER. As mentioned in §3.2, rather than using a product of the style metrics and a single reward token, we can use a *vectorized* reward function that outputs each of the three style metrics individually and correspondingly condition on each of these specific metrics.

Results are shown in Table 4. Incorporating a fine-grained reward improves performance across all dimensions, including $\mathcal{V}$. This shows that conditioning on fine-grained rewards can lead to more control across each desired attribute, resulting in much better style transfers overall.

## 4.7 Analysis of $\mathcal{D}_f$

We analyze $\mathcal{D}_f$, the dataset resulting from the expert-guided dataset generation. First, we compare the lexical diversity of $\mathcal{D}_f$ against existing style transfer corpora. Following Gehrmann et al. (2021), we gauge the mean segmented token-type ratio over segemented length of $N = 10$ (**MSTTR**) and the 1/2/3-gram **entropy** of the training split of each corpus. We also assess the quality of style-transferred outputs in each corpus by assessing fluency (**F**) and meaning similarity (**MS**).

Table 5 shows comparisons of these metrics. The automatically-created $\mathcal{D}_f$ is comparable to existing human-created datasets in diversity and in fluency. The average meaning similarity is also promising, as it is within 85% of the value of GYAFC. This shows the potential of machine-generated data when aided with creative decoding algorithms.

## 5 Related Work

**Style Transfer** Due to the absence of large-scale parallel corpora for text style transfer (TST), prior work has focused on unsupervised methods designed for non-parallel datasets (Dai et al., 2019; Luo et al., 2019). Most of these efforts focus on disentangling the representation of content and the style of a given text, either through an auxiliary discriminator to classify text attributes (Hu et al., 2018; Shen et al., 2017b), or by training with a policy gradient (Xu et al., 2018; Gong et al., 2019).

Recent work has leveraged the generation capabilities of LMs for TST: Krishna et al. (2020), create a pseudo-parallel corpus by paraphrasing text from a style, then training an *inverse* paraphraser to convert text to that style. Other work automatically align pairs of sentences in different styles, either in the representation-level (Prabhumoye et al., 2018) or corpus-level (Liu et al., 2022b).

Others have attempted TST by prompting LMs (Reif et al., 2021; Suzgun et al., 2022). How-

ever, these approaches often rely on a strong initial model, either already fine-tuned on TST-related tasks (e.g. paraphrasing), or a large LM capable of few-shot generalization. In contrast, our framework does not assume strong capabilities of the initial model, making it applicable in a realistic setting.

**RL for NLP**  Recent work has shown the potential of RL to align with arbitrary natural language objective functions across areas such as summarization (Paulus et al., 2017), open-ended text generation (Lu et al., 2022), dialogue (Li et al., 2016; Zhou et al., 2017), question-answering (Liu et al., 2022a), machine translation (Nguyen et al., 2017; Wu et al., 2016), and dataset generation (Pyatkin et al., 2022; Kim et al., 2023). For unified style transfer, a setting where the desired output can be directly correlated to automatic metrics, RL is a promising avenue.

**Data Generation with LMs**  LM-generated data have been increasingly used across a wide range of tasks, such as commonsense reasoning (West et al., 2022; Zelikman et al., 2022), NLI (Ye et al., 2022) and dialogue generation (Kim et al., 2023). While previous approaches rely on the task-solving capability of LLMs, recent work show that small LMs can also generate high-quality datasets without supervision (Jung et al., 2023; Brahman et al., 2023). Building on top of these, our work pushes further on machine-generated data by incorporating 1) inference-time decoding algorithms and 2) targeted filtering, yielding an effective pseudo-parallel corpus to initialize offline reinforcement learning.

## 6  Conclusion

We propose STEER, a unified framework to overcome the challenge of limited parallel data in style transfer, by leveraging expert-guided decoding and two-stage reinforcement learning. We focus on a more realistic use case: rewriting text from an *arbitrary, unknown* style to a desired target style. Through extensive experiments, we demonstrate the effectiveness and robustness of STEER on both in- and out-of-domain style transfer, outperforming competitive baselines. The success of STEER underscores the potential of RL algorithms when combined with controllable decoding and encourages future algorithmic innovation that fully unleash the power of RL for real-world NLP applications.

## 7  Limitations, Ethical Considerations, and Broader Impacts

While STEER demonstrates promising results for arbitrary-to-many style transfer, there are several limitations. Firstly, in our experiments, we rely heavily on the availability of a corpus containing text from diverse styles to act as source styles for the expert-guided creation of $\mathcal{D}_f$; however, not every corpus will have as diverse a set of styles to create a $D_f$ from. Instead, in data-limited settings, it may be required to gather source text from other locations, like other corpora, in order to create candidate style-transfer pairs. Secondly, while we tested the generalization of STEER to out-of-domain source style, adaption to new *target* styles through continual learning requires further investigation and experimentation.

Additionally, like many other natural language systems, STEER could unintentionally introduce harmful stereotypes or engage in malicious content generation. Specially the use of fine-grained reward signals during online training may be used to reinforce undesired behaviors potentially leading to the generation of biased or unethical outputs. Furthermore, bad actors may try to intentionally utilize style transfer systems like STEER to create harm or to harass marginalized communities by using toxic output styles. This is a common misuse case in generation (McGuffie and Newhouse, 2020), and an application which we strongly condemn.

On the positive side, STEER allows for training memory and cost-efficient training of unified style transfer models using existing corpora. Our method is thus beneficial for somewhat reducing the carbon footprint by reducing the reliance on training large language models (LLMs) to achieve desired results (Strubell et al., 2019).

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

# A Modeling Details

## A.1 Out-of-the-Box Modeling

For this work, we use the publicly-available HuggingFace Transformers library (Wolf et al.,

2019), to fine-tune GPT2-models and run inference. The HuggingFace code is available at `https://github.com/huggingface/transformers.` and is licensed under the Apache License 2.0. We also use the OpenAI library (documented at `https://platform.openai.com/docs/libraries`) to make queries to GPT-3 for some of our style transfer baselines.

## A.2 Generation of $\mathcal{D}_f$

We first train GPT2-large experts using the training data from CDS. See Krishna et al., 2020 for more details. For each style expert, we train with the following parameters, shown in Table 6.

| Hyperparameter | Assignment |
|---|---|
| model | GPT2-large |
| number of gpus | 1 |
| effective batch size | 32 |
| total steps | 50,000 |
| steps per evaluation | 1000 |
| learning rate optimizer | AdamW |
| AdamW initial learning rate | 1e-05 |
| AdamW epsilon | 1e-05 |
| learning rate schedule | linear with no warmup |
| weight decay | 0.0 |
| max sequence length | 50 |
| max generation length | 100 |
| padding sequences | to max seq length |

Table 6: Hyperparameters used to finetune the expert models

We also include an early stopping criteria based on repeatedly increasing loss. Each model was trained on a single Nvidia RTX6K with 24GB of memory for less than 24 hours.

After these experts are created, for every pair of styles in CDS, we generate 10,000 *candidate* rewrite pairs. Each source text is randomly selected from the train set of a CDS style, and used as the input text for all other target styles with a variety of hyperparameters. Specifically, we employ the DEXPERTS decoding framework for each of these generations. DEXPERTS has one hyperparameter, $\alpha$, which we vary. We attempt the following hyperparameters for each generation, shown in Table 7:

| Hyperparameter | Tested |
|---|---|
| $\alpha$ | [0.2,0.4,0.6] |
| temperature | [0.7, 1.0, 1.3] |

Table 7: Hyperparameters tested and used for STEER for data over-generation

Besides these hyperparameters, we use the same max sequence and generation length as in §6, and use a no-repeat ngram value of 3 to avoid repetition. After creating $\mathcal{D}_i$, we filter the dataset down to 2.2M rows by doing top-200K sampling for each of the 11 target styles in CDS. Next, we use this data to train the unified model:

## A.3 Training the Unified Model

We train our online RL for 30,000 steps with early stopping based on the overall score $\mathcal{V}$. We use 4 RTX-A100 GPUs with 80GB of memory, and this takes about 120 hours to train. We use a batch size of 256 and the Adam optimizer with learning rate of 1e-5. The full hyperparameters are reported in Table 8.

| Hyperparameter | Assignment |
|---|---|
| model | GPT2-large |
| number of gpus | 4 |
| effective batch size | 1024 |
| total steps | 30,000 |
| steps per evaluation | 2500 |
| learning rate optimizer | AdamW |
| AdamW initial learning rate | 1e-05 |
| AdamW epsilon | 1e-05 |
| learning rate schedule | linear with no warmup |
| weight decay | 0.0 |
| max sequence length | 50 |
| max generation length | 100 |
| padding sequences | to max seq length |
| QUARK hypers | |
| sample steps | 2500 |
| KL penalty | 0.025 |
| entropy coef | 0.00 |
| policy temp. | 1.0 |
| vectorized reward token | True |
| number of buckets | 5 |

Table 8: Hyperparameters used for unified model training, including for QUARK

## B  Dataset Details

### B.1  The Corpus of Diverse Styles (CDS)

CDS is collected and ensembled from online archives and previous academic datasets; we list each style and the train/val/test splits in Table 9. The dataset contains no parallel data, only text from each respective style. We use CDS as the primary dataset for unified style transfer, as it has 11 distinct styles that we can train and evaluate on. See Krishna et al., 2020 for a full explanation of the styles.

| Style | Train | Validation | Test |
|---|---|---|---|
| AAE | 720K | 7.3K | 7.3K |
| Bible | 31K | 1.7K | 1.7K |
| Coha 1810 | 205K | 5.3K | 5.3K |
| Coha 1890 | 1.2M | 32K | 32K |
| Coha 1990 | 1.9M | 49K | 49K |
| English Tweet | 5.2M | 40K | 40K |
| Joyce | 37K | 2.0K | 2.0K |
| Lyrics | 4.5M | 252K | 252K |
| Romantic Poetry | 27K | 1.5K | 1.5K |
| Shakespeare | 25K | 1.3K | 1.3K |
| Switchboard | 146K | 1.5K | 1.5K |

Table 9: Styles and dataset sizes in CDS

## C  Baseline Details

### C.1  Style Transfer via Paraphrasing

We retrieve the diverse paraphraser and the inverse style transfer models from the repository in the original paper; please see Krishna et al., 2020 for more details. At inference, we use greedy decoding as this led to the best results in the original paper.

### C.2  Prompt-and-Rerank

Prompt-and-Rerank has two prompting strategies that do not require knowledge of the source style and are therefore suitable for arbitrary style transfer. The two strategies are vanilla prompting and contrastive prompting. See Reif et al., 2021 for full details on the exact prompts.

We test out both prompts with a small subset of data, and find that contrastive prompting works much better, so we use this going forward. We also try generating $k = 3$ samples and $k = 5$ samples per input, and find that $k = 3$ works the best. Following the original paper, we use nucleus sampling (Holtzman et al., 2019) with $p = 0.9$ and a temperature of 1.0. Finally, we use GPT-2 Large for fair comparison with STEER.

### C.3  GPT-3

We prompt GPT-3 using nucleus sampling (Holtzman et al., 2019) with $p = 0.9$ and a temperature of 1.0. We include further details on zero-shot and few-shot prompting below.

### C.3.1  Zero-shot

We use the following prompt setup for zero-shot style transfer:

```
Rewite the following sentence
into the style of [target style]
  Original Sentence: [original
sentence]
  Rewritten Sentence:
```

### C.3.2  Few-shot

For few-shot style transfer, we randomly show $k$ examples of the target style sampled from the train set, and prepending them before the 0-shot prompt as follows.

```
Here are some examples of
sentences in the style of [target
style]:
  [example 1]
  [example 2]
  [example 3]
  ...
```

## D  Human Evaluation

Since automatic metrics alone have been shown insufficient for evaluating text generations (Novikova et al., 2017), we conduct human evaluation. Annotators rate meaning similarity of a style transfer pair, and the fluency of the style-transferred text.

For fluency, annotators choose between: **0** for *not fluent*, **1** for *somewhat fluent*, and **2** for *fluent*. For meaning similarity, annotators choose between: **0** for *not similar*, **1** for *somewhat similar*, and **2** for *similar*. We discard annotations where all three annotators disagree on a label for either fluency or similarity, resulting in a final human evaluation labeled size of 310 (from an initial size of 330).

To reduce labor cost, we only run our human evaluations on the top three methods from Table 1, meaning we exclude P-A-R. In addition, following previous work, we do not run human evaluation on target style strength. Further details are explained in Appendix D.1.

### D.1  Style Identification Task Difficulty

The target styles in the CDS dataset are extremely complex. Previous work from Krishna et al., 2020 mention that this is too challenging of a task, even for experienced annotators.

We verify the difficulty of the text style identification task reported in Krishna et al., 2020 by performing an additional human evaluation. From the CDS test set, we randomly sample 10 examples from each of the 11 styles (110 total examples with ground truth styles). Next, we use the same three annotators from our previous human evaluation (NLP experts), and provide them with a natural language description of each of the 11 styles and 20 random examples from the train set of each to familiarize them with text from different styles. We ask them to assign a style label to each of the 110

examples, given their knowledge of the styles, and calculate their accuracy and agreement. On average, the annotators only have a 40.0% classification accuracy with an inter-annotator agreement of 0.39 (Fleiss' kappa). In contrast, on the same samples (unseen by the classifier), our classifier obtains a 84.5% classification accuracy. These results validate the difficulty of the task and suggest that an automatic classifier is more suited for this task.

## E   The Cold Start Problem in RL

Reinforcement learning often involves optimizing a **policy** model towards an optimal distribution that *maximizes* some expected reward. This paradigm works well out-of-the-box for a variety of tasks in NLP, such as model detoxification and sentiment control (Lu et al., 2022) where the output distribution of the initial policy *already aligns*, to a reasonable degree, with the optimal reward distribution However, in a cold-start, reinforcement-learning setting, the initial policy output distribution is drastically different than the optimal reward distribution; this may be the case when the reward is linked to a specific task outside the capabilities of the original policy.

Adjusting to cold-start has been mostly explored in the context of recommender systems, where it is difficult to determine user-preferences without any initial data (Ding and Soricut, 2017; Ji et al., 2021; Du et al., 2022), but has been sparsely pursued in reinforcement learning for NLP. Ding and Soricut (2017) introduce *softmax policy gradients* for cold-start reinforcement-learning, but the approach is limited to only one class of reinforcement learning algorithms (policy-gradient approaches) and includes mathematical assumptions not widely applicable to various NLP applications.

## F   Alternative Evaluation Metrics

To ensure that the model improvement from STEER is meaningful (i.e., to make sure our model is not reward hacking), we use a set of alternative metrics for target style strength, meaning similarity, and fluency, and re-run evaluation on all results from Tables 1 and 2. These are metrics *unused* during training time for STEER.

For the fluency model, we use a different binary CoLA classifier (`https://huggingface.co/textattack/roberta-base-CoLA`), and again use the raw probability score of the linguistically acceptable class. To assess meaning

similarity, we use the embedding-based SIM model of Wieting et al., 2019 as used in Krishna et al., 2020. Finally, for the style classifier model, given limited data quantity, we train another RoBERTa-Large classifier with the same CDS data but with a different seed. As before, we compute the aggregate metric $\mathcal{V}$ by taking the product of the three automatic metrics for each style transfer pair in the corpus, and report the average $\mathcal{V}$ value.

Our results using the alternative metrics to re-run evaluation on both in-domain style transfer and out-of-domain style transfer using the CDS-trained STEER mode are shown in Table 10 and Table 11 respectively. Overall, this corroborates our main findings by showing that our relative results are largely unchanged: on the in-domain styles, STEER beats all baselines, including impressive gains on target style strength as well as improved fluency and meaning similarity. On the out-of-domain task, STEER continues to excel, once again beating all other baselines other than GPT-3 on Shakespeare.

## G   Full Experimental Results

We detail the full experimental results for the main experiments in this section, including all style evaluation metrics.

### G.1   Main Experiments

We include the full results for the main experiment from Table 1, testing out style transfer on the CDS dataset to each target style from all other source styles. Table 12 has the results for STEER, Table 13 has the results for STRAP, Table 14 has the results for P-A-R, and Tables 15-18 have the results for GPT-3 0-shot, 1-shot, 5-shot, and 10-shot.

| | GPT-2 Large | | | GPT-3 (`text-davincii-003`) | | | |
|---|---|---|---|---|---|---|---|
| Target Style | **STEER** | STRAP | P-A-R | $k=0$ | $k=1$ | $k=5$ | $k=10$ |
| AAE Twitter | **38.2** | 7.8 | 1.6 | 18.1 | 9.4 | 20.7 | 17.6 |
| Bible | **34.4** | 22 | 2.0 | 4.0 | 12.3 | 14.4 | 13.6 |
| 1810-1820s | **23.1** | 9.2 | 1.0 | 12.8 | 11.5 | 12.3 | 12.6 |
| 1890-1900s | **34.3** | 13.6 | 2.2 | 8.0 | 8.0 | 10.3 | 10.3 |
| 1990-200s | **39.0** | 16.1 | 1.7 | 8.7 | 13.1 | 15.6 | 15.8 |
| English Twitter | **37.7** | 8.7 | 2.5 | 28.2 | 16.6 | 24.2 | 19.8 |
| James Joyce | **17.6** | 10.7 | 1.9 | 1.9 | 1.5 | 1.5 | 2.2 |
| Song Lyrics | **29.3** | 20.1 | 2.9 | 10.3 | 13.6 | 9.4 | 11.7 |
| Romantic Poetry | **16.8** | 14.8 | 0.9 | 0.5 | 2.6 | 4.2 | 3.9 |
| Shakespeare | **11.4** | 8.0 | 0.8 | 5.9 | 7.3 | 6.6 | 6.9 |
| Switchboard | **41.4** | 15.2 | 0.5 | 0.0 | 0.2 | 3.0 | 6.9 |
| **Overall** | **29.4** | 13.3 | 1.6 | 8.9 | 8.7 | 11.1 | 11.0 |

Table 10: Comparison of 11-way style transfer on the CDS dataset measured by aggregate score $\mathcal{V}$ using the alternative evaluation metrics (replication of Table 1). **Bold** and underline denote the highest and the second-highest score respectively in each row.

| | GPT2-Large | | | | | | GPT-3 (`text-davincii-003`) | | | | | | | |
|---|---|---|---|---|---|---|---|---|---|---|---|---|---|---|
| | **STEER** | | STRAP | | P-A-R | | $k=0$ | | $k=1$ | | $k=5$ | | $k=10$ | |
| Target Style | Inf. | For. | Inf. | For. | Inf. | For. | Inf. | For. | Inf. | For. | Inf. | For. | Inf. | For. |
| AAE Twitter | **38.7** | **43.8** | 18.7 | 14.1 | 14.5 | 5.7 | 25.7 | 23.5 | 16.7 | 15 | 25.5 | 24.8 | 23.8 | 22.9 |
| Bible | **29.6** | **32.4** | 19.4 | 20.1 | 0.3 | 0.5 | 3.9 | 3.9 | 10.8 | 11.3 | 13.6 | 15.5 | 15.3 | 15.4 |
| 1810-1820s | **19.6** | **22.3** | 5.2 | 8.4 | 0.2 | 1.5 | 9.9 | 12.6 | 11.2 | 14.1 | 12.6 | 16.7 | 12.1 | 15.7 |
| 1890-1900s | **29.2** | **32.3** | 10.9 | 14.2 | 2.0 | 5.3 | 8.5 | 10.2 | 11.3 | 12.6 | 12.4 | 12.8 | 11.4 | 11.2 |
| 1990-200s | **44.9** | **51.9** | 22.1 | 31.6 | 6.7 | 17.9 | 15.3 | 18.9 | 26.3 | 30.6 | 30.1 | 32.6 | 26.8 | 31.0 |
| English Twitter | **44.4** | **51.2** | 20.6 | 22.1 | 23.6 | 22.5 | 28.0 | 32.3 | 20.3 | 19.8 | 24.7 | 26.0 | 22.8 | 24.5 |
| James Joyce | **19.0** | **21.5** | 11.7 | 13.2 | 1.5 | 3.5 | 2.0 | 2.6 | 2.9 | 3.1 | 2.9 | 2.3 | 3.0 | 3.1 |
| Song Lyrics | **36.4** | **36.1** | 21.3 | 23.6 | 6.0 | 4.3 | 15.3 | 10.0 | 19.0 | 14.6 | 15.7 | 13.3 | 19.2 | 16.1 |
| Romantic Poetry | **11.2** | **11.4** | 10.0 | 11.2 | 0.2 | 0.3 | 1.4 | 0.7 | 3.7 | 2.9 | 4.7 | 3.5 | 4.2 | 2.8 |
| Shakespeare | 10.1 | 9.7 | 9.9 | 9.4 | 0.3 | 1.6 | 10.2 | 12.4 | 11.3 | 12.5 | 11.1 | 12.5 | **11.5** | **12.8** |
| Switchboard | **42.5** | **47.3** | 22.6 | 27.7 | 2.1 | 1.7 | 0.1 | 0.0 | 0.3 | 0.0 | 4.8 | 6.6 | 8.1 | 12.4 |
| **Overall** | **29.6** | **32.7** | 15.7 | 17.9 | 5.2 | 5.9 | 10.9 | 11.6 | 12.2 | 12.4 | 14.4 | 15.1 | 14.4 | 15.3 |

Table 11: Comparison of style transfer to each of the 11 styles in the CDS dataset measured by aggregate score $\mathcal{V}$ using the alternative evaluation metrics from two out-of-domain styles from the GYAFC corpus (replication of Table 2). For. and Inf. denote the formal and informal styles respectively. **Bold** and underline denote the highest and the second-highest score respectively in each row.

| | $\mathcal{V}$ | TSS | F | MS |
|---|---|---|---|---|
| AAE Twitter | 42.6 | 68.2 | 87.0 | 72.5 |
| Bible | 44.0 | 80.9 | 85.9 | 63.9 |
| 1810-1820s | 30.2 | 49.8 | 81.9 | 74.7 |
| 1890-1900s | 35.9 | 56.6 | 85.4 | 74.6 |
| 1990-200s | 42.3 | 63.0 | 83.9 | 77.8 |
| English Twitter | 41.2 | 62.6 | 87.7 | 74.5 |
| James Joyce | 20.4 | 34.1 | 81.8 | 76.7 |
| Song Lyrics | 33.3 | 51.6 | 87.2 | 74.7 |
| Romantic Poetry | 20.4 | 35.3 | 80.5 | 74.7 |
| Shakespeare | 13.6 | 24.7 | 83.7 | 72.7 |
| Switchboard | 52.9 | 92.0 | 87.0 | 66.1 |
| **Overall** | 34.3 | 56.3 | 84.7 | 73.0 |

Table 12: Full results on CDS test set with STEER

| | $\mathcal{V}$ | TSS | F | MS |
|---|---|---|---|---|
| AAE Twitter | 7.4 | 21.8 | 65.7 | 66.2 |
| Bible | 26.9 | 70.5 | 79.4 | 51.7 |
| 1810-1820s | 11.1 | 22.5 | 78.6 | 64.9 |
| 1890-1900s | 12.3 | 22.6 | 82.6 | 65.0 |
| 1990-2000s | 16.6 | 29.1 | 82.8 | 65.0 |
| English Twitter | 8.0 | 36.9 | 76.8 | 49.8 |
| James Joyce | 11.8 | 26.4 | 78.8 | 66.7 |
| Song Lyrics | 20.2 | 39.4 | 80.2 | 67.3 |
| Romantic Poetry | 15.7 | 46.5 | 64.5 | 62.1 |
| Shakespeare | 9.1 | 27.8 | 68.0 | 62.2 |
| Switchboard | 21.1 | 52.0 | 69.3 | 65.2 |
| **Overall** | 14.6 | 36.0 | 75.2 | 62.4 |

Table 13: Full results on CDS test set with the STRAP

|  | $\mathcal{V}$ | TSS | F | MS |
|---|---|---|---|---|
| AAE Twitter | 3.8 | 10.3 | 76.1 | 68.5 |
| Bible | 6.6 | 24 | 79.1 | 67.3 |
| 1810-1820s | 3.5 | 8.7 | 74.4 | 78.9 |
| 1890-1900s | 4.4 | 10 | 76 | 77.7 |
| 1990-2000s | 4.3 | 9.1 | 75.4 | 79.2 |
| English Twitter | 5.5 | 12.5 | 74.1 | 82.1 |
| James Joyce | 5.4 | 13.6 | 76.9 | 75.3 |
| Song Lyrics | 7.7 | 23.2 | 78.2 | 65.3 |
| Romantic Poetry | 2.8 | 7.6 | 77.1 | 76.1 |
| Shakespeare | 2.5 | 8.2 | 77 | 72 |
| Switchboard | 1.7 | 3.9 | 75.1 | 84.5 |
| **Overall** | 4.4 | 11.9 | 76.3 | 75.2 |

Table 14: Full results on CDS test set with PAR

|  | $\mathcal{V}$ | TSS | F | MS |
|---|---|---|---|---|
| AAE Twitter | 23.2 | 58.4 | 63.1 | 65.7 |
| Bible | 5.2 | 10.4 | 85 | 70.4 |
| 1810-1820s | 14.7 | 27.6 | 79.4 | 67 |
| 1890-1900s | 8.6 | 15 | 83.1 | 65 |
| 1990-2000s | 7.9 | 12.6 | 83.8 | 68.8 |
| English Twitter | 35 | 58.3 | 82 | 70 |
| James Joyce | 3.4 | 7.4 | 77.1 | 66 |
| Song Lyrics | 12.2 | 25.4 | 73.7 | 72.2 |
| Romantic Poetry | 1.1 | 3.5 | 61.7 | 61.5 |
| Shakespeare | 9.6 | 27.7 | 64.3 | 68.6 |
| Switchboard | 0.1 | 0.1 | 92.2 | 74.5 |
| **Overall** | 11 | 22.4 | 76.9 | 68.2 |

Table 15: Full results on CDS test set for GPT-3 0-shot.

|  | $\mathcal{V}$ | TSS | F | MS |
|---|---|---|---|---|
| AAE Twitter | 11.2 | 30.9 | 67.3 | 64.8 |
| Bible | 16 | 40.2 | 79 | 59.5 |
| 1810-1820s | 15.9 | 29.4 | 82 | 65.3 |
| 1890-1900s | 9.1 | 16 | 82.4 | 63.3 |
| 1990-2000s | 13 | 25.2 | 86.2 | 65.3 |
| English Twitter | 23.6 | 47 | 71.3 | 68 |
| James Joyce | 1.3 | 11 | 46.1 | 63.2 |
| Song Lyrics | 15.4 | 40.1 | 71.8 | 62.4 |
| Romantic Poetry | 3.4 | 15.3 | 51.7 | 59.1 |
| Shakespeare | 10 | 29.5 | 61.9 | 69 |
| Switchboard | 0.3 | 0.6 | 90.3 | 75.4 |
| **Overall** | 10.8 | 25.9 | 71.8 | 65 |

Table 16: Full results on CDS test set for GPT-3 1-shot.

|  | $\mathcal{V}$ | TSS | F | MS |
|---|---|---|---|---|
| AAE Twitter | 25.4 | 79.1 | 58.4 | 59 |
| Bible | 20.2 | 43.6 | 78.8 | 65.6 |
| 1810-1820s | 17.4 | 35.6 | 80.4 | 62.4 |
| 1890-1900s | 10.4 | 17.2 | 84.6 | 66.8 |
| 1990-2000s | 17.5 | 26.9 | 89.1 | 71 |
| English Twitter | 32 | 55.9 | 80.2 | 68.4 |
| James Joyce | 1.6 | 10.9 | 48.6 | 63.2 |
| Song Lyrics | 11.2 | 25.8 | 74.7 | 66 |
| Romantic Poetry | 6.2 | 28.5 | 44.6 | 57.4 |
| Shakespeare | 9.7 | 28 | 62.8 | 68.4 |
| Switchboard | 5.3 | 10.7 | 85.4 | 71.2 |
| **Overall** | 14.3 | 32.9 | 71.6 | 65.4 |

Table 17: Full results on CDS test set for GPT-3 5-shot.

|  | $\mathcal{V}$ | TSS | F | MS |
|---|---|---|---|---|
| AAE Twitter | 22.7 | 65.4 | 57.9 | 64.9 |
| Bible | 21 | 43.8 | 81.4 | 65.7 |
| 1810-1820s | 17 | 33.1 | 81 | 66.2 |
| 1890-1900s | 10.1 | 16.9 | 83.1 | 68.8 |
| 1990-2000s | 17.2 | 25.7 | 88.2 | 72.1 |
| English Twitter | 29.5 | 51.4 | 77.8 | 69.9 |
| James Joyce | 2.6 | 20.9 | 33.9 | 62.4 |
| Song Lyrics | 13.2 | 30.8 | 69.5 | 69.5 |
| Romantic Poetry | 4.9 | 23.2 | 43.9 | 58.9 |
| Shakespeare | 9.7 | 28.7 | 60.8 | 69.5 |
| Switchboard | 13.7 | 24.8 | 82.2 | 73.5 |
| **Overall** | 14.7 | 33.2 | 69.1 | 67.4 |

Table 18: Full results on CDS test set for GPT-3 10-shot.