# OpenReview forum: "STEER: Unified Style Transfer with Expert Reinforcement"
_EMNLP/2023/Conference — EMNLP 2023 Findings_

### Official Review · Reviewer_kv8D · 2023-07-26

**Typos Grammar Style And Presentation Improvements:** 1. Citations should be added to the s…
**Soundness:** 3

**Excitement:**

3: Ambivalent: It has merits (e.g., it reports state-of-the-art results, the idea is nice), but there are key weaknesses (e.g., it describes incremental work), and it can significantly benefit from another round of revision. However, I won't object to accepting it if my co-reviewers champion it.

**Paper Topic And Main Contributions:**

In this work, the authors propose a unified framework for arbitrary style transfer. It first generates pseudo data using expert language models, then filters these data based on a score function that takes style, fluency, and semantic meaning into consideration, and finally leverages a reinforcement learning algorithm named Quark to train the model. The authors propose to replace the scalar reward with a vectorized reward function. Experiments show that the proposed framework outperforms the compared baselines significantly measured by a specific score.

**Questions For The Authors:**

1. Do I understand correctly that the equation in line 226 is used as the evaluation metric?
2. Any comments on reasons to reject 1 and 2? I will raise my score if they are addressed properly.
3. What metrics do the subfigures 2~4 in Figure 3 correspond to?
4. How is the value $\mathcal{V}$ in Table 4 calculated? I can not obtain it using the equation in line 226, i.e., $0.428\times0.626\times0.790 \ne 0.1994$.

**Reasons To Accept:**

1. The framework does not rely on large language models but achieves good results.
2. Experiments are conducted on an 11-style dataset, making the results more solid.

**Reasons To Reject:**

1. If I understand correctly, the authors leverage the equation in line 226 as the evaluation metric. However, the metric is directly used to filter data and used as the reward function in reinforcement learning. In other words, this framework is optimized directly against the evaluation metric but the baselines are not, which may be not fair.
2. The evaluation metric used in this work is different from highly related previous works (Suzgun et al., 2022; Krishna et al., 2020), but the reason is not explained.

[1] Mirac Suzgun, Luke Melas-Kyriazi, and Dan Jurafsky. 2022. Prompt-and-rerank: A method for zero-shot and few-shot arbitrary textual style transfer with small language models. In Proceedings of the 2022 Conference on Empirical Methods in Natural Language Processing, pages 2195–2222, Abu Dhabi, United Arab Emirates. Association for Computational Linguistics.
[2] Kalpesh Krishna, John Wieting, and Mohit Iyyer. 2020. Reformulating unsupervised style transfer as paraphrase generation. In Proceedings of the 2020 Conference on Empirical Methods in Natural Language Processing (EMNLP), pages 737–762, Online. Association for Computational Linguistics.

**Reproducibility:**

4: Could mostly reproduce the results, but there may be some variation because of sample variance or minor variations in their interpretation of the protocol or method.

**Reviewer Confidence:**

4: Quite sure. I tried to check the important points carefully. It's unlikely, though conceivable, that I missed something that should affect my ratings.

---

> ### Author Rebuttal · Authors · 2023-08-29
>
> Thank you for your detailed and specific feedback! We were happy to read that you appreciated our main points: that STEER achieves “good” performance on a diverse dataset which makes our results more “solid”, and that our method results in a model outperforming the much larger Instruct GPT-3 model, showing that we “[do] not rely on large language models”.
>
> Please see our response below. We would love to address additional questions during the discussion period if anything is unclear.
>
> ---
>
> **STEER is optimized directly against the evaluation metric but the baselines are not, which may be unfair**
>
> Defining a task-specific reward function is a common practice in the Reinforcement Learning (RL) field, and a standard procedure adopted across various RL literature (Lu et al., 2022; Ramamurthy et al., 2022; Wu et al., 2023). To this end, we use reinforcement learning to closely, directly align our style transfer model with the desired properties (measured by the evaluation metrics) by using a fine-grained reward. In fact, the baseline we included, Suzgun et al., 2022, devises the same strategy: **reranking** and choosing the best style transfer from multiple candidates based on the evaluation metrics is an important part of the pipeline.
>
> We additionally verify the quality of generations through a human evaluation on meaning similarity and fluency of style transfer outputs (line 413, and Figure 4). We do not run human evaluation on target style strength following Krishna et al., 2020, given the difficulty of the task with so many target styles. Human evaluation results corroborate our automatic metrics, showing that STEER outperforms baselines on the partially human aggregate metric (the product of the human evaluation fluency and meaning similarity with the automatic evaluation of target style strength).
>
> Finally, we re-run the numbers from Table 1 using alternative fluency, meaning similarity, and target style strength models to give a fair comparison to other baselines, and ensuring that STEER is not optimized on the evaluation metrics. For the fluency model, we use a different binary CoLA classifier (https://huggingface.co/textattack/roberta-base-CoLA), and again use the raw probability score of the linguistically acceptable class. To assess meaning similarity, we use the embedding-based SIM model of Wieting et al., 2018 as used in Krishna et al, 2020. Finally, for the style classifier model, given limited data quantity, we train another RoBERTa-Large classifier with the same CDS data but with a different seed. **We recompute the three metrics for both in-domain and out-of-domain style transfer on CDS** (replicating Tables 1 and 2) and the aggregate metric $\mathcal{V}$, finding that **our relative results are largely unchanged**: on the in-domain styles, STEER beats all baselines, including impressive gains on target style strength as well as improved fluency and meaning similarity. On the out-of-domain task, STEER continues to excel, once again beating all other baselines other than GPT-3 on Shakespeare. The absolute metrics for fluency and $\mathcal{V}$ are slightly lower across all methods due to the alternate fluency classifier. We will include these results in the next version of the paper.
>
> ---
>
> **The evaluation metric used in this work is different from highly related previous works (Suzgun et al., 2022; Krishna et al., 2020), but the reason is not explained.**
>
> Our evaluation is in fact **very similar to previous work** from Krishna et al., 2020 and Suzgun et al., 2022 with slight modifications. We explain our differences below:
>
> Krishna et al., 2020 assesses overall style transfer quality using a fluency model (RoBERTa trained on the CoLA corpus (Warstadt et al, 2018), a style transfer accuracy model (11-way RoBERTa classifier), and a semantic similarity model (subword embedding-based SIM model of Wieting et al., 2018). For each style transfer input/output, they also take the product of the three three metrics to create an **aggregate** metric (sentence-level aggregation). Importantly, they **binarize** the style transfer accuracy and fluency based on a threshold of 0.5 before this product.
>
> We follow this evaluation pipeline closely, using the same fluency and style accuracy model, with two modifications designed to enable more fine-grained quality evaluation: 1) Rather than binarizing style transfer accuracy and fluency when calculating the sentence-level aggregate metric V, we use the **raw probability scores** from the respective style classifier and fluency models, a float between 0-1 each. As in Reif et al., 2021, this allows us to represent the **strength** of the target style (as well as fluency), which gives us more fine-grained evaluation and better distinguishes the quality of similar style transfer pairs. 2) Rather than using the SIM model (Wieting et al., 2019), we use Sentence-BERT (Reimers and Gurevych, 2019) since it is more widely-used (5k vs 100 citations), is tested on more benchmarks, and is easier to implement.
>
> **Overall, we closely follow the evaluation from Krishna et al., 2020**. Our minor changes are designed to improve evaluation and are backed by prior work. We will clarify this and include these justifications in the next version of the paper.
>
> ---
>
> **Question: Do I understand correctly that the equation in line 226 is used as the evaluation metric?**
>
> Correct. We use the aggregate metric of target style strength, meaning similarity, and fluency to evaluate the quality of our models. We also report a full breakdown of the three individual metrics in the **Appendix, Tables 9-15** corresponding to all the methods to Table 1, which shows that our method, STEER, increases the performance of **all** metrics over baselines almost all of the time. In addition, we conduct a human evaluation to ensure the quality of the style transfer outputs (line 413, and Figure 4).
>
> ---
>
> **Question: What metrics do the subfigures 2~4 in Figure 3 correspond to?**
>
> This corresponds to the style transfer metrics we define on lines 200-228: target style strength, fluency, meaning similarity, and the product of these metrics, $\mathcal{V}$, an aggregate metric.  ($\mathcal{V}$ is the same metric reported in Tables 1 and 2). Figure 3 shows the results of an ablation where we vary the initial data pool used in the reinforcement learning procedure.
>
> ---
>
> **Question: How is the value in Table 4 calculated? I can not obtain it using the equation in line 226**
>
> We calculate the aggregate metric $\mathcal{V}$ on the **sentence level** (following previous work from Krishna et al., 2020). Each style transfer input/output has the following associated metric values: target style strength (TSS), meaning similarity (MS), output fluency (F), and the product of these metrics, $\mathcal{V}$ (line 226, as you mentioned). In Table 4, we report the **average** of these metrics across all sentences in the CDS test set. Though for each sentence $\mathcal{V}$ is equal to the product of the TSS, MS, and F, once we take the corpus-average of these metrics, we lose this equality guarantee. **$\mathcal{V}$ in Table 4 is a *average of products* while taking the product of TSS, MS, and F (from Table 4) as you suggested would be a *product of averages*.**
>
> ---
>
> **Typos, Grammar, Style, And Presentation Improvements**
>
> Thanks for the suggestions! We will correct these in the next version of this work, including adding a citation on arbitrary style transfer to the second paragraph of the introduction, correcting the typos, and making the evaluation metrics more clear.
>
> ---
>
> **References**
>
> Krishna, K., Wieting, J., & Iyyer, M. (2020). Reformulating Unsupervised Style Transfer as Paraphrase Generation. ArXiv, abs/2010.05700.
>
> Lu, X., Welleck, S., Jiang, L., Hessel, J., Qin, L., West, P., Ammanabrolu, P., & Choi, Y. (2022). Quark: Controllable Text Generation with Reinforced Unlearning. ArXiv, abs/2205.13636.
>
> Ramamurthy, R., Ammanabrolu, P., Brantley, K., Hessel, J., Sifa, R., Bauckhage, C., Hajishirzi, H., & Choi, Y. (2022). Is Reinforcement Learning (Not) for Natural Language Processing?: Benchmarks, Baselines, and Building Blocks for Natural Language Policy Optimization. ArXiv, abs/2210.01241.
>
> Reif, E., Ippolito, D., Yuan, A., Coenen, A., Callison-Burch, C., & Wei, J. (2021). A Recipe for Arbitrary Text Style Transfer with Large Language Models. ArXiv, abs/2109.03910.
>
> Reimers, N., & Gurevych, I. (2019). Sentence-BERT: Sentence Embeddings using Siamese BERT-Networks. Conference on Empirical Methods in Natural Language Processing.
>
> Suzgun, M., Melas-Kyriazi, L., & Jurafsky, D. (2022). Prompt-and-Rerank: A Method for Zero-Shot and Few-Shot Arbitrary Textual Style Transfer with Small Language Models. ArXiv, abs/2205.11503.
>
> Warstadt, A., Singh, A., & Bowman, S.R. (2018). Neural Network Acceptability Judgments. Transactions of the Association for Computational Linguistics, 7, 625-641.
>
> Wieting, J., Berg-Kirkpatrick, T., Gimpel, K., & Neubig, G. (2019). Beyond BLEU:Training Neural Machine Translation with Semantic Similarity. Annual Meeting of the Association for Computational Linguistics.
>
> Wu, Z., Hu, Y., Shi, W., Dziri, N., Suhr, A., Ammanabrolu, P., Smith, N.A., Ostendorf, M., & Hajishirzi, H. (2023). Fine-Grained Human Feedback Gives Better Rewards for Language Model Training. ArXiv, abs/2306.01693.

---

### Official Review · Reviewer_kVMX · 2023-08-03

**Soundness:** 3

**Excitement:**

3: Ambivalent: It has merits (e.g., it reports state-of-the-art results, the idea is nice), but there are key weaknesses (e.g., it describes incremental work), and it can significantly benefit from another round of revision. However, I won't object to accepting it if my co-reviewers champion it.

**Paper Topic And Main Contributions:**

This paper proposes a novel approach STEER for text style transfer allowing transfer from a wider range of source styles. Their contribution focuses on an expert-guided pseudo-parallel data generation and a reinforcement learning training strategy adapted from Quark. They use language models finetined on each style dataset as experts and anti-experts to modulate the output logits of the data generation LLM following product of experts. These generated data serve as the initial data pool for reinforcement learning after top-K filtration. The training is alternatively updating the reward model and data pool. Experimental results on 3 datasets with at most 11 styles demonstrate the proposed STEER has significant advantage over previous methods.

**Reasons To Accept:**

* This work proposes a new way to overcome the lack of parallel data, pushing style transfer closer to real-world applications.
* The improvement is large compared with previous methods.

**Reasons To Reject:**

* This work still does not realize transferring from *arbitrary unknown* styles since data generation needs stylistic data as input. Ideally, the data is supposed to be constructed unsupervisedly. The performance on out-of-domain styles is questionable.
* I don’t understand why in the human evaluation the authors use an automatic metric TSS rather than a human metric to evaluate the style control. This weakens the convincingness of human evaluation.
* The trend of NLP is towards zero-shot and few-shot. The approaches requiring heavy finetuning are less convenient in practice.

**Reproducibility:**

4: Could mostly reproduce the results, but there may be some variation because of sample variance or minor variations in their interpretation of the protocol or method.

**Reviewer Confidence:**

3: Pretty sure, but there's a chance I missed something. Although I have a good feel for this area in general, I did not carefully check the paper's details, e.g., the math, experimental design, or novelty.

---

> ### Author Rebuttal · Authors · 2023-08-29
>
> Thank you for taking the time to review our paper! We were happy to read that you appreciated our main points, including our strong experimental results demonstrating a “[large] improvement” over baselines, and that our method helps overcome the common limitation in style transfer of the lack of parallel data, which is “pushing style transfer closer to real-world applications”.
>
> Please see our response to your questions and concerns below. We would love to address additional questions during the discussion period if anything is unclear.
>
> ---
> **This work still does not realize transferring from arbitrary unknown styles since data generation needs stylistic data as input. Ideally, the data is supposed to be constructed unsupervisedly. The performance on out-of-domain styles is questionable.**
>
> We would like to clarify our approach - while our method does **initially** require data where the style is known to create an initial data pool (expert-guided data generation), the resulting model trained with this diverse dataset can transfer text both from styles present in the initial data pool, and from **out-of-domain, unseen** styles.  In other words, the resulting model can successfully transfer from any **arbitrary** style. In Table 2 (also lines 403-412), we show performance of the STEER model trained on the CDS corpus transferring from two out-of-domain (OOD) styles. The two OOD styles are unseen by the model and are fed into the CDS-trained model **without additional data generation or model training**. STEER outperforms all baselines transferring to each of the 11 styles in the CDS dataset from the two OOD styles, except to GPT3 on Shakespeare. We will make our out-of-domain experiments and results more clear in the next version of the paper.
>
> ---
>
> **Unclear why authors do not perform human evaluation for target style strength**
>
> As discussed in **Footnote 6 & Appendix E, line 1047**, we follow previous work (Krishna et al, 2020; authored the CDS dataset) in not performing human evaluation on target style transfer accuracy, as this is a challenging task especially for untrained annotators who are unfamiliar with the set of target styles.
>
> We additionally verify the difficulty of this task by performing a human evaluation. From the CDS test set, we randomly sample 10 examples from each of the 11 styles (110 total examples with ground truth styles). Next, we use the same three annotators from our previous human evaluation (NLP experts), and provide them with a natural language description of each of the 11 styles and 20 random examples from the train set of each to familiarize them with text from different styles. We ask them to assign a style label to each of the 110 examples, given their knowledge of the styles, and calculate their accuracy and agreement. On average, the **annotators only have a 40.0% classification accuracy** with an inter-annotator agreement of 0.39 (Fleiss’ kappa). In contrast, on the same samples (unseen by the classifier), our **classifier obtains a 84.5% classification accuracy**. These results validate the difficulty of the task and suggest that an automatic classifier is more suited for this task.
>
> ---
>
> **The trend of NLP is towards zero-shot and few-shot. The approaches requiring heavy finetuning are less convenient in practice.**
>
> While zero-shot and few-shot methods are promising (Brown et al., 2020; Suzgun et al., 2022), we would like to emphasize that our method has significant merits (1) in making **smaller, more accessible models** available for wider adaptation and broader community usage, and (2) **achieving better or comparable results** than much larger and not widely accessible models, both of which are increasingly important considerations in NLP research (Strubell et al., 2019; Schwartz, 2019; Schick et al., 2020). Also, we would like to clarify that STEER only requires a one-time fine-tuning step.
>
> More specifically, STEER outperforms prompting baselines (Table 1 and 2), on **all in-domain styles**, and on all **out-of-domain** styles, except for GPT3 on Shakespeare, by a significant margin. This is especially impressive as the STEER model is based on GPT2-Large, a 774M parameter model, while the Instruct-GPT3 baseline has 175B parameters, a 226 times disparity in model size. While prompting for style transfer is also possible with smaller models (Suzgun et al., 2022), we find the performance of prompting methods with smaller models (using the P-A-R method from Suzgun et al., 2022 with GPT2-Large, results shown in Table 1 and 2) to be **extremely poor, especially transferring to lesser known/more obscure styles** such as the switchboard corpus.
>
> Additionally, STEER only requires a **one-time fine-tuning step** which equips it with style transfer capability from any **arbitrary out-of-domain** source style to several target styles (see Table 2 for specific results). This further enhances the convenience of STEER as a single unified model that can be used for unseen inputs.
>
> ---
>
> **References**
>
> Brown, T.B., Mann, B., Ryder, N., Subbiah, M., Kaplan, J., Dhariwal, P., Neelakantan, A., Shyam, P., Sastry, G., Askell, A., Agarwal, S., Herbert-Voss, A., Krueger, G., Henighan, T.J., Child, R., Ramesh, A., Ziegler, D.M., Wu, J., Winter, C., Hesse, C., Chen, M., Sigler, E., Litwin, M., Gray, S., Chess, B., Clark, J., Berner, C., McCandlish, S., Radford, A., Sutskever, I., & Amodei, D. (2020). Language Models are Few-Shot Learners. ArXiv, abs/2005.14165.
>
> Krishna, K., Wieting, J., & Iyyer, M. (2020). Reformulating Unsupervised Style Transfer as Paraphrase Generation. ArXiv, abs/2010.05700.
>
> Schick, T., & Schütze, H. (2020). It’s Not Just Size That Matters: Small Language Models Are Also Few-Shot Learners. ArXiv, abs/2009.07118.
>
> Schwartz, R., Dodge, J., Smith, N., & Etzioni, O. (2019). Green AI.
>
> Strubell, E., Ganesh, A., & McCallum, A. (2019). Energy and Policy Considerations for Deep Learning in NLP. ArXiv, abs/1906.02243.
>
> Suzgun, M., Melas-Kyriazi, L., & Jurafsky, D. (2022). Prompt-and-Rerank: A Method for Zero-Shot and Few-Shot Arbitrary Textual Style Transfer with Small Language Models. ArXiv, abs/2205.11503.

---

### Official Review · Reviewer_3tNE · 2023-08-05

**Typos Grammar Style And Presentation Improvements:** None come to mind
**Soundness:** 3

**Excitement:**

3: Ambivalent: It has merits (e.g., it reports state-of-the-art results, the idea is nice), but there are key weaknesses (e.g., it describes incremental work), and it can significantly benefit from another round of revision. However, I won't object to accepting it if my co-reviewers champion it.

**Missing References:**

None come to mind

**Paper Topic And Main Contributions:**

This paper introduces a model that is capable of transferring multiple source styles into multiple target styles by a) first creating a pseudo parallel dataset using the combination of a paraphraser model which can be steered or guided to produce the target style using signals from the product of experts mechanism and b) uses this new dataset D_f as a supervised training problem to train an initial RL policy. Finally c) in online training, this initial policy is used for generation to augment the dataset while also updating the policy alternatively to improve the model's performance in a typical RL fashion. Finally, they show that it also performs well when unseen source-type sentences are introduced.

The authors claim that this model beats strong baselines significantly through their experiments primarily by arguing that their model on average produces the highest augmented score V across all datasets compared to even larger models like davinci 003. Some side studies are done to show that a vectorised signal consisting of fluency, STS and content sim. work better during RL training.



**Questions For The Authors:**

1. Can you address the questions posed above related to why the human evaluations skew more towards gpt3.
2. The question related to if you think the augmented score V is a true indicator of style transfer quality.

**Reasons To Accept:**

1. This model shows the efficacy of RL to train and finetune conditional generative models. Though it has been explored before, the combination of creating a supervised dataset to train the initial policy and then subsequently improving the policy during the online training seems novel. This might be a helpful signal to future work in conditional generation (not specifically arbitrary style transfer).
2. Considering a many-to-many style transfer problem seems like the right direction to pursue rather than similar work that focuses on a one-to-one task. The authors have used sufficient datasets to prove the model's effectiveness for this task.
3. The experiments are well-defined and the explanations are sufficient to get a clear idea of what the model does and how it does it without sacrificing brevity and ease of understanding. The structure of the people can be improved a little bit, however (e.g. analysis of D_f in 4.6 can be added earlier).
4. Overall this word does have merits both conceptually and through experimental results. However, one cannot say for sure that concepts from this work will be re-used in this setting, particularly in style transfer, as they are heavily engineered.

**Reasons To Reject:**

1. The major issue with this work is the claim that it performs better than strong baselines such as gpt3. For example, in Figure 4 showing the results of human evaluation, we can see that gpt3 performs better than steer in meaning similarity and fluency. Whereas steer outperforms gpt3 in only style transfer strength. This indicates that steer might suffer from the inherent flaw that most models do generally in this problem i.e. gaining style transfer strength while compromising on content similarity. It is quite hard to gauge if steer indeed does perform better than gpt3 given these conflicting results.
2. One might argue that your augmented score definition V consisting of a linear product of fluency, content sim and style transfer metrics offers a singular score to judge a model. Therefore according to Table 2, steer clearly seems like the best-performing model. However, an adversarial situation can be thought of where a model greedily optimises and outperforms in style transfer acc. by a large degree compared to the baselines but sacrifices on content sim., enough to push the augmented score V as high as possible (this is a well-studied issue in the style transfer task where it is still an open problem).
3. Apart from the two above points, the other consideration is that this model inherently needs two-phase training (granted it performs well in unseen future styles). There are more intermediate steps such as dataset filtering that are also needed. Prompt and LLMs do not require any supervision and therefore are even more generalised approaches for style transfer, although they are indeed more computationally expensive.
4. To summarise, it is not convincing that steer is the best model due to the conflects between the automated metrics and human evaluations.


**Reproducibility:**

3: Could reproduce the results with some difficulty. The settings of parameters are underspecified or subjectively determined; the training/evaluation data are not widely available.

**Reviewer Confidence:**

4: Quite sure. I tried to check the important points carefully. It's unlikely, though conceivable, that I missed something that should affect my ratings.

---

> ### Author Rebuttal · Authors · 2023-08-29
>
> Thank you for your detailed and specific feedback! We were happy to read that you appreciated our main points: that our two-stage process to improve a mediocre initial policy is “novel” and could be a “helpful signal to future work in conditional generation”, that our strong experimental results have “prove[d] the model's effectiveness”, and that our work has “merits both conceptually and through experimental results”.
>
> Please see our response to your valuable questions and feedback below. We would love to address additional questions during the discussion period if anything is unclear.
>
> ---
>
> **It is hard to tell if STEER performs better overall than GPT3 since on human evaluation, STEER outperforms GPT3 on style transfer strength, but not on fluency and meaning similarity**
>
> Thanks for bringing this up! Prior work demonstrates a **noticeable tradeoff between style transfer accuracy and meaning preservation**; this is often shown **experimentally** (Suzgun et al., 2022; Wu et al., 2019; Malmi et al., 2020; Wu et al., 2019; Xu et al., 2019; Li et al., 2018) and also **explicitly discussed** (Li et al., 2018; Wu et al., 2019; Xu et al., 2019; Hallinan et al., 2023). Intuitively, when transferring from one style to another, some amount of semantic changes is unavoidable; as a simple example, meaning similarity will be maximized when the input is naively copied.
>
> Due to this tradeoff between style transfer accuracy and meaning preservation, it is often impossible to completely optimize for both metrics, and different methods inherently have different advantages. In this case, the **scale of the difference of the metrics** in Figure 4 between STEER and GPT3 is important: STEER has an impressive **44% higher target style strength** while sacrificing **24% decrease in meaning similarity** and a **negligible 1.9% decrease in fluency**. Overall, we think this is a reasonable tradeoff, as STEER sacrifices much less fluency and meaning preservation for much more style transfer accuracy.
>
> Finally, one advantage of STEER is we can **explicitly control the tradeoff** between different style transfer objectives (style transfer accuracy, meaning similarity, fluency) by altering the weights assigned to the metrics when computing the aggregate reward in both the **data generation** and **reinforcement learning** step. This allows one to promote specific qualities in the outputs; for example, we can encourage preserving meaning more by assigning a higher weight to meaning similarity relative to target style strength. We will highlight this advantage more explicitly in our next revision.
>
> ---
>
> **Though STEER obtains a high aggregate V score, it may be greedily optimizing for target style strength at the sacrifice of fluency and meaning similarity**
>
> That’s indeed a potential concern, and one which we considered and were able to avoid! We list the breakdown of automatic metrics (target style strength, meaning similarity, fluency, and the aggregate metric $\mathcal{V}$) for all methods in Table 1 in the **Appendix, Tables 9-15**.
>
> Comparing Table 9, which has the full metrics results from STEER on the CDS test set, to Tables 10-15 (results of other baselines), shows that our target style strength does indeed significantly improve over baselines while also showing that our meaning similarity and fluency improve almost all of the time. This demonstrates that **STEER does not optimize only for target style strength, and in fact improves all three metrics jointly**, resulting in better overall style transfers. In the rare cases in which either fluency or meaning similarity are lower than baseline methods, they are still competitive, while having a much higher target style strength. As an example, for the Bible style, STEER gets a target style strength, fluency, and meaning similarity of 80.9, 85.9, 63.9 respectively; while GPT3 0-shot for the same style has a higher meaning similarity of 70.4, the target style strength is only 10.4 (the fluency is similar at 85.0).
>
> Overall, these results demonstrate that in addition to strong target style strength, our high aggregate metric can be attributed to better or competitive fluency and meaning similarity (ie, we're not optimizing only the target style strength).
>
> ---
>
> **Prompting approaches with unsupervised LLMs may be more generalizable because STEER requires a two-phase training**
>
> While zero-shot and few-shot methods are promising (Brown et al., 2020; Suzgun et al., 2022), we would like to emphasize that our method has significant merits (1) in making **smaller, more accessible models** available for wider adaptation and broader community usage, and (2) **achieving better or comparable results** than much larger and not widely accessible models, both of which are increasingly important considerations in NLP research (Strubell et al., 2019; Schwartz, 2019; Schick et al., 2020). Also, we would like to clarify that STEER only requires **a one-time fine-tuning step**.
>
> More specifically, STEER outperforms prompting baselines (Table 1 and 2), on **all in-domain styles**, and on all **out-of-domain** styles, except for GPT3 on Shakespeare, by a significant margin. This is especially impressive as the STEER model is based on GPT2-Large, a 774M parameter model, while the Instruct-GPT3 baseline has 175B parameters, a 226 times disparity in model size. While prompting for style transfer is also possible with smaller models (Suzgun et al., 2022), we find the performance of prompting methods with smaller models (using the method from Suzgun et al, 2022 with GPT2-Large, results shown in Table 1 and 2) to be **extremely poor, especially transferring to lesser known/more obscure styles** such as the switchboard corpus.
>
> Additionally, STEER only requires **a one-time fine-tuning step** which equips it with style transfer capability from any **arbitrary out-of-domain** source style to several target styles (see Table 2 for specific results). This further enhances the convenience of STEER as a single unified model that can be used for unseen inputs.
>
> ---
>
> **Can you address the questions posed above related to why the human evaluations skew more towards gpt3.**
>
> Human evaluations in meaning similarity may skew toward **GPT3** because it is **known to perform well on paraphrasing and summarization** as shown in multiple literature (Brown et al., 2020; Goyal et al., 2023; Zhang et al., 2023). However, we demonstrate that it **lacks the skill of successfully transferring to styles**. As shown above, though GPT3 outperforms STEER on human evaluation meaning similarity and fluency, STEER performs much better on target style strength.
>
> ---
>
> **Is the augmented score V a true indicator of style transfer quality?**
>
> We follow prior work in style transfer, which suggests aggregating the style transfer accuracy, meaning similarity, and fluency metrics (Krishna et al., 2020; Xu et al., 2018; Pang and Gimpel, 2018; Li et al., 2018). This ensures that all three metrics are optimized and enables a singular comparison between different methods (Krishna et al., 2020). In addition, we also report individual metrics in the Appendix (as discussed above) and conduct human evaluation (lines 413-444) to assess the true quality of the style transfers.
>
> ---
>
> **One cannot say for sure that concepts from this work will be re-used in this setting, particularly in style transfer, as they are heavily engineered.**
>
> Although our method requires engineering to initially build the model, we think the concepts will be reused both for the taks of arbitrary style transfer and in other potential settings.
>
> As mentioned above, STEER requires only an **initial** data generation and model training to create a model that can transfer from **arbitrary source styles** to different target styles (see Table 2 for results). Though there is an initial engineering cost to create this model, this is a **one-time** cost and leads to **a capable and robust** downstream model.
>
> Additionally, the concepts from our method are general and can be potentially applied to other tasks. Specifically, in reinforcement learning, using offline data exploration to supplement an insufficient initial policy could be useful in scenarios where there is a sparse reward signal or where there is a lack of initial, high-quality data, such as in style transfer for styles lacking parallel data.
>
> ---
>
> **References**
>
> Brown, T.B., Mann, B., Ryder, N., Subbiah, M., Kaplan, J., Dhariwal, P., Neelakantan, A., Shyam, P., Sastry, G., Askell, A., Agarwal, S., Herbert-Voss, A., Krueger, G., Henighan, T.J., Child, R., Ramesh, A., Ziegler, D.M., Wu, J., Winter, C., Hesse, C., Chen, M., Sigler, E., Litwin, M., Gray, S., Chess, B., Clark, J., Berner, C., McCandlish, S., Radford, A., Sutskever, I., & Amodei, D. (2020). Language Models are Few-Shot Learners. ArXiv, abs/2005.14165.
>
> Goyal, T., Li, J.J., & Durrett, G. (2022). News Summarization and Evaluation in the Era of GPT-3. ArXiv, abs/2209.12356.
>
> Hallinan, S., Liu, A., Choi, Y., & Sap, M. (2022). Detoxifying Text with MaRCo: Controllable Revision with Experts and Anti-Experts. Annual Meeting of the Association for Computational Linguistics.
>
> Krishna, K., Wieting, J., & Iyyer, M. (2020). Reformulating Unsupervised Style Transfer as Paraphrase Generation. ArXiv, abs/2010.05700.
>
> Li, J., Jia, R., He, H., & Liang, P. (2018). Delete, Retrieve, Generate: a Simple Approach to Sentiment and Style Transfer. North American Chapter of the Association for Computational Linguistics.
>
> Malmi, E., Severyn, A., & Rothe, S. (2020). Unsupervised Text Style Transfer with Masked Language Models. EMNLP.
> Pang, R., & Gimpel, K. (2018). Unsupervised Evaluation Metrics and Learning Criteria for Non-Parallel Textual Transfer. Conference on Empirical Methods in Natural Language Processing.
>
> Schick, T., & Schütze, H. (2020). It’s Not Just Size That Matters: Small Language Models Are Also Few-Shot Learners. ArXiv, abs/2009.07118.
>
> Schwartz, R., Dodge, J., Smith, N., & Etzioni, O. (2019). Green AI.
>
> Strubell, E., Ganesh, A., & McCallum, A. (2019). Energy and Policy Considerations for Deep Learning in NLP. ArXiv, abs/1906.02243.
>
> Suzgun, M., Melas-Kyriazi, L., & Jurafsky, D. (2022). Prompt-and-Rerank: A Method for Zero-Shot and Few-Shot Arbitrary Textual Style Transfer with Small Language Models. ArXiv, abs/2205.11503.
>
> Wu, X., Zhang, T., Zang, L., Han, J., & Hu, S. (2019). Mask and Infill: Applying Masked Language Model for Sentiment Transfer. IJCAI.
>
> Xu, J., Sun, X., Zeng, Q., Ren, X., Zhang, X., Wang, H., & Li, W. (2018). Unpaired Sentiment-to-Sentiment Translation: A Cycled Reinforcement Learning Approach. Annual Meeting of the Association for Computational Linguistics.
>
> Xu, R., Ge, T., & Wei, F. (2019). Formality Style Transfer with Hybrid Textual Annotations. ArXiv, abs/1903.06353.
>
> Zhang, T., Ladhak, F., Durmus, E., Liang, P., McKeown, K., & Hashimoto, T. (2023). Benchmarking Large Language Models for News Summarization. ArXiv, abs/2301.13848.

---

### Meta-Review · Area_Chair_us3N · 2023-09-18

**Recommendation:** 4

**Metareview:**

Using Reinforcement Learning (RL), this work proposes a method for training generative style-transfer models that is particularly useful in settings with limited parallel data. Additionally, this work demonstrates out-of-domain generalization and shows improvement over large language models (LLMs) such as GPT-3, despite being significantly smaller in size. One main concern raised by reviewers is the comparatively complicated method required to train the model, especially when compared to the simplicity of prompting or zero-shot with LLMs. However, I would argue that this isn't really a fair comparison. While LLMs offer the convenience of prompting, they are also computationally expensive. In many cases, fine-tuned, smaller models should outperform zero-shot prompting, as is the case here.

Another point raised by reviewers is whether the evaluation adequately captures general improvement. While STEER has a superior aggregate score, it's important to note that it excels significantly in terms of style strength at the expense of meaning similarity. Therefore, the utility of STEER may depend on whether preserving the original meaning is a requirement or not.

---

### Decision · Program_Chairs · 2023-10-07

**Decision:**

Accept-Findings

**Comment:**

Using Reinforcement Learning (RL), this work proposes a method for training generative style-transfer models that is particularly useful in settings with limited parallel data. Additionally, this work demonstrates out-of-domain generalization and shows improvement over large language models (LLMs) such as GPT-3, despite being significantly smaller in size. One main concern raised by reviewers is the comparatively complicated method required to train the model, especially when compared to the simplicity of prompting or zero-shot with LLMs. However, I would argue that this isn't really a fair comparison. While LLMs offer the convenience of prompting, they are also computationally expensive. In many cases, fine-tuned, smaller models should outperform zero-shot prompting, as is the case here.

Another point raised by reviewers is whether the evaluation adequately captures general improvement. While STEER has a superior aggregate score, it's important to note that it excels significantly in terms of style strength at the expense of meaning similarity. Therefore, the utility of STEER may depend on whether preserving the original meaning is a requirement or not.